# CRISPR/Cas12a Coupling with Magnetic Nanoparticles and Cascaded Strand Displacement Reaction for Ultrasensitive Fluorescence Determination of Exosomal miR-21

**DOI:** 10.3390/molecules27165338

**Published:** 2022-08-22

**Authors:** Qing Liu, Jingjian Liu, Na He, Moli Zhang, Lun Wu, Xiyu Chen, Jun Zhu, Fengying Ran, Qinhua Chen, Hua Zhang

**Affiliations:** 1Oncology Department, Fujian Medical University Union Hospital, Fujian Provincial Key Laboratory of Translational Cancer Medicine, Fuzhou 350001, China; 2Sinopharm Dongfeng General Hospital, Hubei University of Medicine, Shiyan 442008, China; 3Shenzhen Baoan Authentic TCM Therapy Hospital, Shenzhen 518101, China

**Keywords:** fluorescence, miRNA-21, CRISPR/Cas12a, strand displacement reaction, magnetic nanoparticles

## Abstract

Exosomal MicroRNA-21 (miRNA-21, miR-21) is significantly up-regulated in blood samples of patients with lung cancer. Exosomal-derived miR-21 can be used as a promising biomarker for the early diagnosis of lung cancer. This paper develops a fluorescent biosensor based on the combination of magnetic nanoparticles (MNPs), cascade strand displacement reaction (CSDR) and CRISPR/Cas12a to detect the exosomal miR-21 from lung cancer. The powerful separation performance of MNPs can eliminate the potential interference of matrix and reduce the background signal, which is very beneficial for the improvement of specificity and sensitivity. The CSDR can specifically transform one miR-21 into plenty of DNA which can specifically trigger the trans-cleavage nuclease activity of Cas12a, resulting in the cleavage of ssDNA bi-labeled with fluorescent and a quencher. Under the optimized experimental conditions, the developed fluorescence biosensor exhibited high sensitivity and specificity towards the determination of exosomal-derived miR-21 with a linear range from 10 to 1 × 10^5^ fM and a low detection limit of about 0.89 fM. Most importantly, this method can be successfully applied to distinguish the exosomal miR-21 from the lung cancer patients and the healthy people.

## 1. Introduction

Cancer ranks as a leading cause of death and an important barrier to increasing life expectancy in every country of the world [1]. In 2020, lung cancer was the second most common cancer and the main cause of cancer death, accounting for about 1 in 10 cancers diagnosed and 1 in 5 deaths [2,3]. According to statistics, the 5-year survival rate of patients with early lung cancer is significantly higher than that of patients with advanced lung cancer [4]. Therefore, the early diagnosis of lung cancer is particularly important. Liquid biopsy has been recognized as one of the most promising strategies for the early diagnosis of cancers, and various bodily fluids, such as blood, urine and saliva, can be used as the objects of liquid biopsy [5]. The biomarkers that are highly associated with tumor growth and metastasis, including circulating tumor DNA (ctDNA), circulating tumor RNA (ctRNA), extracellular vesicles (EVs), and circulating tumor cells (CTCs), can be used as indicators for the early diagnosis of cancer [6]. Exosomes are vesicular bodies with a diameter of 30–200 nm, and they are the most promising detection objects in liquid biopsy [7]. Exosomes contain biological molecules such as RNAs, DNAs, MicroRNAs (miRNAs, miRs) and proteins, which play an extremely important role in regulating the occurrence and development of tumors [8]. Compared with free miRNAs, exosome-derived miRNAs have better stability because of the protection of exosomal phospholipid bilayers [9]. Studies have shown that exosomal miRNA-21 (miR-21) is significantly up-regulated in blood samples of patients with lung cancer, which can be used as a biomarker for the early diagnosis of lung cancer [10,11,12]. However, due to the low abundance of exosomal miR-21 in early lung cancer and the complexity of blood components, it still remains challenging to accurately determine the exosomal miR-21 at the early stage of lung cancer.

Reverse transcription-quantitative polymerase chain reaction (RT-qPCR) is often used as the gold-standard technique for the detection of miRNA [13,14]. However, this technology has some disadvantages, such as time-consuming workflow, expensive equipment and so on, which limit their wide application in resource-limited areas. Therefore, biosensors based on electrochemistry [15], fluorescence [16], surface-enhanced Raman scattering [17], and surface plasmon resonance [18] have been developed to detect miRNAs. These biosensors have the advantages of simple operation, high sensitivity, and specificity. Among them, fluorescence stands out from these biosensors for detecting miRNAs because of its simple operation and rapid response [19]. However, the fluorescence biosensor for detecting exosomal miR-21 in early stages of lung cancer in patients has insufficient specificity and sensitivity. Recently, the clustered regularly interspaced short palindromic repeats (CRISPR)-CRISPR associated proteins (Cas) system has attracted attention for its powerful performance in the field of gene editing. In addition, the powerful trans-cleavage activity of the CRISPR/Cas system, which is specifically triggered by the binding of target-crRNA, expands the application of the CRISPR/Cas system as a powerful bioanalytical tool [20,21]. As a member of the CRISPR/Cas system family, the Cas12a system can bind with the single-strand DNA (ssDNA) or double-strand DNA (dsDNA) without the PAM (protospacer adjacent motif), thus triggering its trans-cleavage activity [22]. Thence, the CRISPR/Cas12a-based biosensors usually exhibit high specificity toward the detection of target [23].

However, the limitations of the CRISPR/Cas12a-based fluorescence biosensor need to be overcome to enable the determination of the ultra-low abundance miR-21. It is necessary to introduce signal amplification strategy. Various signal amplification strategies including duplex-specific-nuclease (DSN) [24], rolling circle amplification (RCA) [25], hybridization chain reaction (HCR) [26] and strand displacement reaction (SDR) had been applied in biosensing of exosomal miR-141 [27]. The SDR can specifically transform the input of miR-21 to the output of ssDNA, which can bind with the crRNA of CRISPR/Cas12a, resulting in the cleavage of the fluorescent reporter, which is bi-labeled with fluorescence and a quencher. In addition, the cascade strand displacement reaction (CSDR) can realize the “one miR-21 into plenty of DNA” signal amplification. The constant coexistence of reactants and products in the same solution results in high background signal, which hinders the improvement of the sensitivity and specificity of the biosensor. Magnetic nanoparticles (MNPs) can be easily separated from the background matrix under a magnetic field. Therefore, the introducing MNPs can eliminate the possibility that the trans-cleavage activity of CRISPR/Cas12a was triggered in the absence of miR-21. Thence, the MNPs is very beneficial for improving the sensitivity and specificity of biosensor. As far as we know, the fluorescence biosensor based on the combination of MNPs, CSDR and CRISPR/Cas12a for the determination of exosomal-derived miRNA has not been reported.

In this work, a CRISPR/Cas12a-based fluorescence biosensor combined with MNPs and CSDR has been developed for the determination of exosomal miR-21 derived from lung cancer. The miR-21 can switch on the CSDR, resulting in a large amount of probe 3 (P3) releasing from the MNPs. After the magnetic separation, the free P3 binds with the crRNA of CRISPR/Cas12a, resulting in the trans-cleavage activity of CRISPR/Cas12a being triggered. The fluorescent reporter which was bi-labeled with fluorescence and a quencher (ssDNA-FQ probe) was cleaved, leading to the generation of fluorescence signal. By combining the advantages of the target-specifical triggered trans-cleavage activity of CRISPR/Cas12a, the “one miR-21 into plenty of DNA” signal amplification of CSDR and the powerful separation ability of MNPs, the developed fluorescence biosensor can enable the determination of miR-21 with ultra-high sensitivity and specificity. The fluorescent biosensor can successfully distinguish the exosomal miR-21 from lung cancer patients and healthy people, showing the potential of this assay in the analysis of real samples.

## 2. Materials and Methods

### 2.1. Materials

HPLC-purified oligonucleotides were synthesized and purified by Sangon Biotechnology Co. Ltd. (Shanghai, China, www.sangon.com, accessed on 6 June 2021). All the nucleic acid sequences used in this work are listed in Figure 1. Streptavidin-modified MNPs (SA-MNPs, 40 nm) were bought from XFNANO Co., Ltd. (Nanjing, China, www.xfnano.com, accessed on 6 June 2021). EnGen^®^ Lba Cas12a was purchased from New England Biolabs Ltd. (Beverly, MA, USA). Phosphate-buffered saline (PBS) was bought from Sigma-Aldrich (USA). The other reagents were of analytical grade and used without further purification. Ultrapure water obtained from a Millipore water purification system (18.2 MΩ·cm resistivity, Milli-Q Direct 8) was used in all runs. Human plasma was obtained from the Sinopharm Dongfeng General Hospital, Hubei University of Medicine, and approved by the Sinopharm Dongfeng General Hospital’s Ethics Committee (Shiyan, China).

### 2.2. Extraction and Purification of Exosomes

The patients were clinically diagnosed as having lung cancer, and their blood samples were centrifuged at 300× *g* for 10 min to remove cells and obtain plasma. First, the plasma (1 mL) was sequentially centrifuged at 4 °C at 300× *g*, 2000× *g*, and 10,000× *g* for 10 min, 20 min, and 30 min to remove cell debris and other biomacromolecules completely. Then, after ultracentrifugation at 100,000× *g* for 70 min, the supernatant was discarded. The sediment was resuspended in 20 mL of sterile phosphate-buffered saline (PBS) and ultracentrifuged at 100,000× *g* for 70 min again. Finally, the exosomes were suspended at 200 μL PBS and stored at −80 °C.

### 2.3. Characterization of Exosomes

#### 2.3.1. Transmission Electron Microscopy

Transmission electron microscopy (TEM) was used to observe the morphology of exosomes. The exosome sample was prepared by the negative-staining method. Firstly, 20 μL the exosome suspension was dripped on the copper grid (200 mesh) and dried at room temperature; then, 20 μL 2% phosphotungstic acid solution (pH 6.8) (Sigma-Aldrich, St. Louis, MO, USA www.sigmaaldrich.com, accessed on 11 August 2021) was added to the copper grid to negative stain at room temperature for 1 min, and the excess negative-staining solution was absorbed with the filter paper. Finally, the copper grid was photographed under the TEM (Hitachi H-7000 NAR) and the morphology of exosomes was observed.

#### 2.3.2. Nanoparticle Tracking Analysis

Nanoparticle tracking analysis (NTA) was applied to analyse the concentration and size distribution of exosomes. The exosome samples were diluted 100-fold with PBS and then determinated on the NanoSight NS300 (Malvern Panalytical Ltd., Malvern, UK). The Brownian motion data of nanoparticles in 1 min were collected and analyzed by NTA2.3 software. The concentration and size distribution of exosomes in each sample were calculated. Each sample was determinated three times.

### 2.4. Extraction and Purification of Total RNAs

ExoRNeasy Midi/Maxi Kit (QIAGEN, www.qiagen.com, accessed on 10 June 2021) was used to purify total RNAs from exosomes. Firstly, 700 μL QIAzol was added to 140 μL extracted exosome solution, centrifuged at 5000× *g* for 5 min, and then incubated for 5 min. Secondly, we added 90 μL chloroform, incubated for 2–3 min, and centrifuged at 12,000 g for 15 min. The upper aqueous phase was transferred to a new collecting tube and twice the 2 vol of 100% ethanol was added. Thirdly, 700 μL sample was centrifuged with RNeasy MinElute spin column 10,000× *g* for 15 s, 700 μL buffer RWT 10,000 g for 15 s, 500 μL buffer RPE 10,000 g for 15 s and 2 min, respectively. Finally, an RNeasy MinElute spin column was run at full speed for 5 min, 14 μL RNase-free water was added directly to the center of the spin column membrane, and the column was left to stand for 1 min and then centrifuged at full speed for 1 min.

### 2.5. Determination of Exosomal miR-21

The detection of exosomal miR-21 was carried out according to the following steps. Firstly, biotin-Probe 1 (P1, 20 nM), Probe 2 (P2, 20 nM) and Probe 3 (P3, 20 nM) were mixed for 20 min. Then, the SA-MNPs solution (20 μg mL^−1^) was added into the solution and reacted upon shaking for 30 min. After magnetic separation, the supernatant was discarded and the magnetic division was dissolved in PBS to obtain the biotin-P1/P2/P3/SA-MNPs. Then, Probe 4 (P4, 30 nM) and miR-21 were added into the above solution. After a period of time, the solution was magnetic separated and the ssDNA-FQ probe (30 nM) and Cas12a/crRNA (60 nM) were added to the supernatant. Forty minutes later, the fluorescence intensity of solution was measured. Each step of the above reaction was magnetically separated and washed three times with PBS.

### 2.6. Fluorescence Measurements

The fluorescence measurements were carried out at room temperature using a Hitachi F-4600 spectrophotometer (Hitachi Co., Ltd. Tokyo, Japan, https://www.hitachi.co.jp, accessed on 6 September 2018) equipped with a xenon lamp excitation source. The excitation wavelength was set at 485 nm, and the fluorescence spectrum was collected at the emission wavelength of 510 to 600 nm. The optimal experimental parameters were studied by using the fluorescence intensity at 518 nm, and the performance of the biosensor was evaluated. The excitation and emission intervals were set at 5 nm. Except for the special group, the fluorescence intensity of each group was measured three times and the standard deviation was calculated. The quantitative expression of exosomal miR-21 was shown by the final fluorescence intensity.

## 3. Results

### 3.1. The Design and Mechanism

Figure 2 shows the schematic illustration of the fluorescent biosensor for exosomal miR-21 determination by CSDR combined with CRISPR/Cas12a. Firstly, P2 and P3 combine with P1 to form the complex of biotin-P1/P2/P3 because of complementary base pairing. The complex of biotin-P1/P2/P3 binds with SA-MNPs to form the biotin-P1/P2/P3/SA-MNPs because of the special combination of biotin and streptavidin. In the presence of miR-21, it will trigger the first round of SDR and replace P3 to form biotin-P1/P2/miR-21/SA-MNPs. P3 is replaced by miR-21 and forms several new unpaired bases, which favor entropy-driven binding between P4 and P1. Thence, the P4 triggers the second round SDR to form the complex of biotin-P1/P4/SA-MNPs resulting in the release of P2 and miR-21. Then, the release miR-21 initiates the next SDR. Thence, one miR-21 can generate plenty of P3 being released from the biotin-P1/P2/P3/SA-MNPs via this SDR. Moreover, the free P3 can bind with the Cas12a/crRNA and the trans cleavage activity of Cas12a is activated to cleave the ssDNA-FQ probe, generating the fluorescence signal. Thence, the concentration of miR-21 can be measured by measuring the fluorescence intensity.

### 3.2. Characterization of Exosomes

Exosomes with complete morphology and function are the critical precondition of related research. The exosomes extracted from human plasma were characterized by TEM and NTA. As shown in Figure 3a, TEM showed that the diameter of the extracted exosomes was about 100 nm and showed typical morphology with a round-like, double-membrane structure. As shown in Figure 3b, the concentration and particle size distribution of exosomes were characterized by NTA. The nanoparticle size of exosomes ranged from 50 to 200 nm, and the concentration was about 3.2 × 10^10^ particles mL^−1^, indicating that the exosome sample had a concentrative size distribution and high purity.

### 3.3. Optimization of the Assay Conditions

To obtain optimal determination performance, the experimental conditions of the fluorescent biosensor were optimized by controlling the following parameters: (a) concentration of MNPs; (b) reaction time of CSDR; (c) concentration of P1; (d) concentration of P4; (e) concentration of Cas12a; (f) reaction time of Cas12a; (g) concentration of ssDNA-FQ probe. The detection performances of biosensor under different parameters were evaluated by fluorescence intensity. As shown in Figure 4, under the following conditions, the performance of biosensor was the best: (a) the optimal concentration of MNPs, 20 μg mL^−1^; (b) the optimal reaction time of CSDR, 30 min; (c) the optimal concentration of P1, 20 nM; (d) the optimal concentration of P4, 30 nM; (e) the optimal concentration of Cas12a, 60 nM; (f) the optimal reaction time of Cas12a, 40 min; (g) the optimal concentration of ssDNA-FQ probe, 30 nM.

### 3.4. Sensitivity

Taking the above optimal parameters as experimental conditions, the performance of fluorescent biosensor was evaluated by different concentrations of miR-21. Figure 5a showed that the fluorescence intensity increased with the increase in miR-21 concentration. As shown in Figure 5b, there was a good linear relationship between the logarithm of concentration of miR-21 with 10 as the base (lgC) and fluorescence intensity. The regression equation was y = 141.7112 × lgC − 0.8841 (R^2^ = 0.9955), where y was the fluorescence intensity and C was the concentration of miR-21. According to the 3σ rule, the limit of detection (LOD) is evaluated to be about 0.89 fM (LOD = 3σ/Slope), and the linear range was from 10 to 1 × 10^5^ fM. In order to further evaluate the performance of this developed biosensor, it was compared with the sensors that had been reported for detecting miR-21. As shown in Table 1, our biosensor had a lower detection limit and wider linear range, especially compared with the other fluorescence-based biosensors.

### 3.5. Specificity

There was a complex nucleic acid environment in the body, and the fluorescent biosensor needed to face the interference of other nucleic acids in specific applications. In order to study the specificity of this biosensor for detecting exosomal miR-21, several nucleic acids similar to miR-21 had been designed, such as SM (single base mismatch), DM (double base mismatch), TM (three base mismatch), and random. As shown in Figure 6, although the concentration of other substances was 10 times that of miR-21, the fluorescent biosensor had the highest fluorescence signal intensity when detecting miR-21, which indicated that the fluorescent biosensor had excellent specificity.

### 3.6. Application

In order to study the performance of the fluorescent biosensor in real samples, it was applied to the detection of exosomal miR-21 in lung cancer patients, and normal human plasma samples were used as the control. After the extraction and identification of exosomes in plasma samples, this fluorescent biosensor was used to detect miR-21 in the extracted total RNA, and the detection procedure was carried out following the above description. As shown in Figure 7a, the fluorescence intensity of exosomal miR-21 in the plasma of patients with lung cancer was significantly stronger than that of the control group composed of normal people. The difference in fluorescence intensity between the lung cancer patients and normal people is statistically significant (*p* < 0.0001). As shown in Figure 7b, the fluorescent biosensor showed excellent detection performance, and the results have good unity with RT-qPCR. This fluorescent biosensor could not only be applied to the detection of exosomal miR-21 in plasma samples, but the detection results were also consistent with the actual situation, which showed that it had good application potential in the detection of clinical samples.

## 4. Conclusions

Herein, a fluorescent biosensor for detecting exosomal miR-21 was designed. The developed fluorescent biosensor makes full use of the synergistic advantages of CRISPR/Cas12a, CSDR and MNPs. The CSDR can specifically transform one miR-21 input to plenty of DNA output, which can trigger the trans-cleavage activity of CRISPR/Cas12a. This process not only converts the detection of miR-21 into the fluorescent signal response but also amplifies the signal response. The MNPs can fully separate the products with reactants. This can further improve the specificity and sensitivity of the biosensor. Thence, the developed fluorescent biosensor exhibits excellent performance in the detection of exosomal miR-21. Even when the concentration of other substances is 10 times that of miR-21, the signal is weaker than that of miR-21. This method also successfully distinguishes the exosomal miR-21 in lung cancer patients from healthy people, which shows its potential in the diagnosis of lung cancer. In addition, the results of this developed method for the determination of exosomal miR-21 from the clinical samples have high consistency with the RT-qPCR. This detection strategy can also be applied to the detection of other biomarkers and provide some help for the monitoring of clinical diseases.

## Figures and Tables

**Figure 1 molecules-27-05338-f001:**
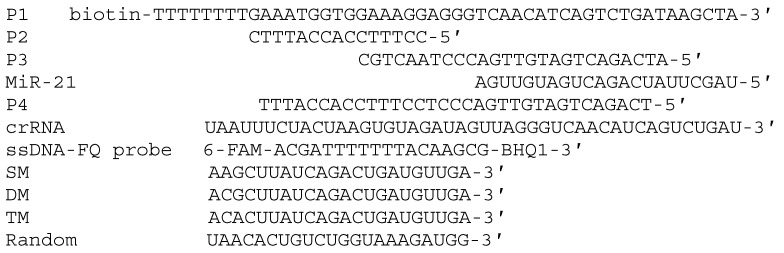
Sequences of nucleic acid used in the experiment.

**Figure 2 molecules-27-05338-f002:**
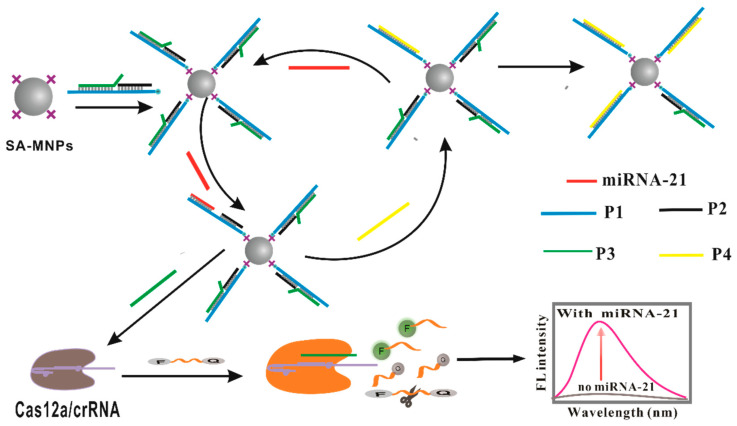
Schematic illustration of fluorescent biosensor for ultrasensitive fluorescence determination of exosomal miR-21 by CRISPR/Cas12a coupling with MNPs and CSDR.

**Figure 3 molecules-27-05338-f003:**
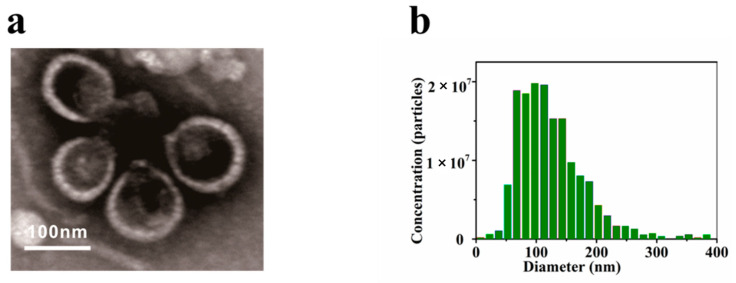
Characterization of exosomes: (**a**) TEM image of exosomes; (**b**) NTA analysis of exosomes.

**Figure 4 molecules-27-05338-f004:**
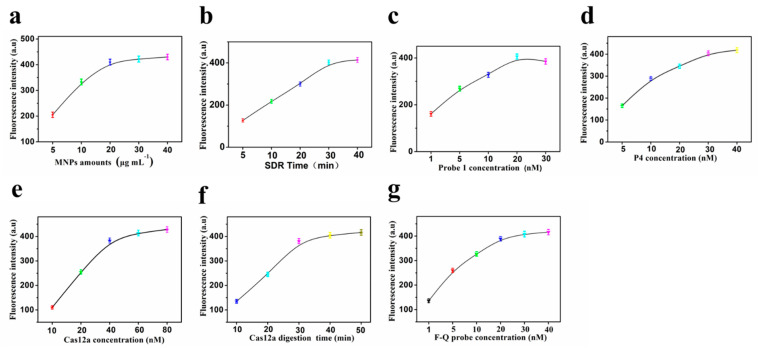
Effects of (**a**) concentration of MNPs; (**b**) reaction time of SDR; (**c**) concentration of P1; (**d**) concentration of P4; (**e**) concentration of Cas12a; (**f**) reaction time of Cas12a; (**g**) concentration of ssDNA-FQ probe. (Error bars: SD, n = 3).

**Figure 5 molecules-27-05338-f005:**
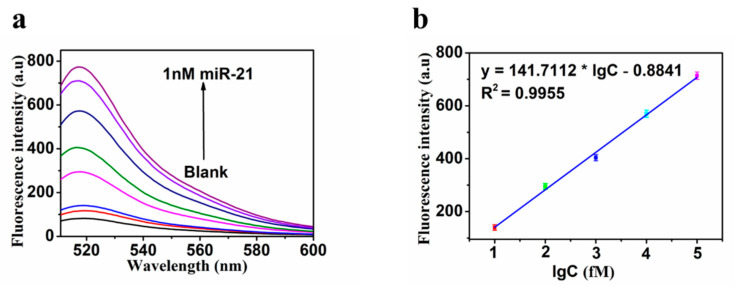
(**a**) Fluorescence emission spectra of the biosensor in the presence of miR-21 with different concentrations (0, 1 fM,10 fM, 100 fM, 1 pM, 10 pM, 100 pM, 1 nM); (**b**) fluorescence intensity as a function of miR-21 concentration. Error bars: SD, n = 3.

**Figure 6 molecules-27-05338-f006:**
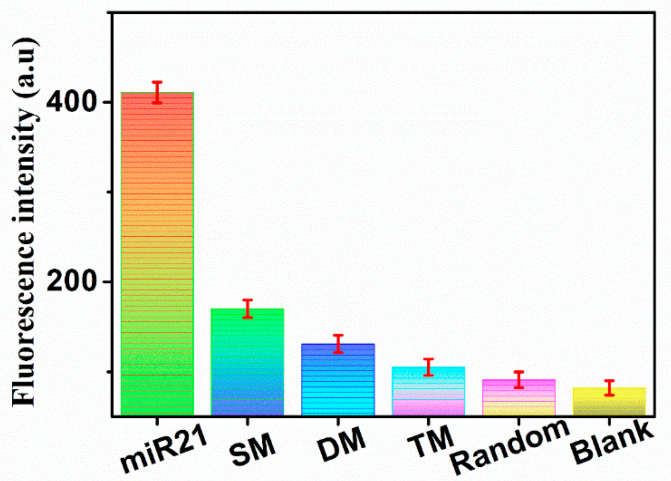
Fluorescence intensity of the fluorescent biosensor in the presence of exosomal miR-21 (1 pM), SM (single-base mismatch, 10 pM), DM (double-base mismatch, 10 pM), TM (triple-base mismatch, 10 pM), random (10 pM), and blank, respectively. (Error bars: SD, n = 3).

**Figure 7 molecules-27-05338-f007:**
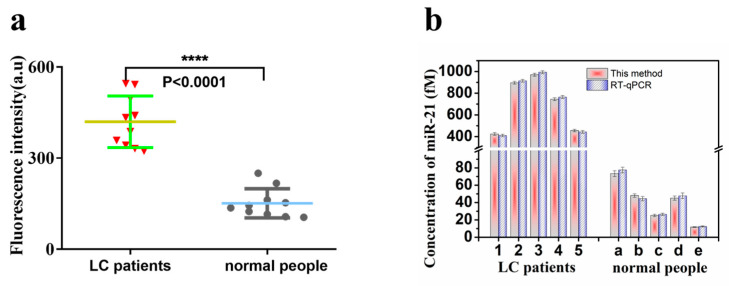
(**a**) A fluorescence biosensor for the detection of exosomal miR-21 from lung cancer patients and normal people (**** *p* < 0.0001) (error bars: SD, n = 10). (**b**) The expression of exosomal miR-21 was detected by the fluorescence biosensor and RT-qPCR, respectively (error bars: SD, n = 3).

**Table 1 molecules-27-05338-t001:** Comparison of miR-21 detection with the reported methods.

Analytical Methods	Detection Limit	Linear Range	Ref.
Quartz crystal microbalance	3.6 pM	2.5 pM to 2.5 μM	[28]
Quartz crystal microbalance	0.87 pM	1.0 pM to 1.0 mM	[29]
Colorimetric	1 pM	1 pM to 1 nM	[30]
Electrochemistry	2.3 fM	10 to 70 fM	[31]
Electrochemistry	26 fM	100 fM to 100 nM	[32]
Electrochemistry	0.01 fM	0.01 fM to 1 μM	[33]
Electrochemistry	0.713 fM	20 fM to 600 pM	[34]
Fluorescence	200 pM	200 pM to 20 nM	[35]
Fluorescence	0.02 nM	0.02 to 5 nM	[36]
Fluorescence	0.89 fM	10 to 1 × 10^5^ fM	This method

## Data Availability

Data are contained within the article.

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
