# Peer review of "CRISPR/Cas12a Coupling with Magnetic Nanoparticles and Cascaded Strand Displacement Reaction for Ultrasensitive Fluorescence Determination of Exosomal miR-21"

_molecules, 2022, doi:10.3390/molecules27165338_

Round 1

Reviewer 1 Report

The authors present a fluorescence-based detection technique for exosomal miR-21. The idea is interesting and the experiments successfully demonstrate the application of the method. I have a few minor comments as follows.

1.       The manuscript should be thoroughly checked once for spelling and grammatical mistakes. For example,

·         Page 1, line 18: This paper develop >> this paper develops

·         Page 2, line 54: it still remain>>it still remains

·         Page 2, line 75: the shortcoming/limitations

·         Page 2, line 86: replace “unbeneficial….” With “interferes in ….”

·         Page 2, line 87: can be easily separated

·         … please correct all such mistakes throughout the whole manuscript.

·         Page 5, line 187: Complete the sentence after “..entropy-driven.”

2.       Why the excitation wavelength was set at 320 nm? FAM excitation is usually around 450-500 nm.

3.       Figure 3: Increase the size of figure labels and overall size. Too small to understand.

If these are taken care of, I recommend the manuscript for publication.

Reviewer 2 Report

This paper reports the application of CRISPR/Cas12a trans‐cleavage activity to detect miR‐21 in exososomal samples with the goal to differentiate patient with lung cancer from healthy people. Exosomes were purified and characterized by TEM. Assay conditions were optimized and sensitivity and specificity was estimated.

Assay’s fluorescence was quantified in 10 lung cancer and healthy people and shown a significant increase in lung cancer samples.

It would be important to compare the sensitivity and specificity of the approach with the gold standard RT‐qPCR and what would the data with healthy and lung cancer samples look like using the gold standard approach to which you must compare, as to me, the gold standard approach is not that much more complicated or needing of much more expensive equipment. 

What was the blood sample volume used to isolate the exosomes?

What was the volume of exosomes and final volume of the reaction used for the assay detection? Given that only concentrations are reported, it is hard to estimate the actual sensitivity of the approach.

The plasma exosome levels can differ in different stages of lung cancer. Could you report the quantification of exosomes in healthy vs lung cancer patients and did you take a fixed volume or a fixed exosome number to perform the quantifications?

It would be nice to show an alignment of the sequences of the different primers used such as (I hope the formatting will not be lost here, but you should get the idea…):

P1 biotin‐TTTTTTTTGAAATGGTGGAAAGGAGGGTCAACATCAGTCTGATAAGCTA‐3’

P2             CTTTACCACCTTTCC‐5’

P3                           CGTCAATCCCAGTTGTAGTCAGACTA‐5’

Mir21                                        AGUUGUAGUCAGACUAUUCGAU‐5’

P4      TTTACCACCTTTCCTCCCAGTTGGTAGTCAGACT‐5’ crRNA UAAUUUCUACUAAGUGUAGAUAGUUAGGGUCAACAUCAGUCUGAU‐3’

If am not mistaken, there is no sequence specificity in the ss‐DNA and the probe is treated as “collateral ssDNA trans‐cleavage”? The design and rationale of the probe should be better explained and Figure 1 should be improved.P4 and mir‐21 in this figure should be better differentiated (not the same color or a more different color)

The manuscript would greatly benefit from English editing. Missing words, bad verb tenses and poor sentence structure all distract and at time confuse the manuscript.

Round 2

Reviewer 2 Report

1.Regarding the RT-qPCR quantification supplementation in the manuscript, please describe with the usual level of details the methodological approach that was used. Also, the analysis provided is only bare minimum, although it surprises me that such a perfect correlation between the two methods could be obtained. I did not see any mention about the RT-qPCR sensitivity and specificity which would provide a much greater analysis.

2.The blood volume used for exosome purification should be added to the manuscript.

3.Please clarify the fact that a fixed volume was used and what was the fraction of the initial 1 ml blood sample, ie, based on your previous answer, it appears to me that all of the purified exosomes obtained from 1 mL was used for the assessment, correct me and clarify if I am wrong about this. Also, I believe this raises the possibility that the increased miR-21 levels detected in the patients may be simply due to increased exosome concentration in the plasma and thus, one may only need to quantify the exosomes and this may lead to as good a biomarker as the miR-21 quantification from exosomes. I think it would be worth to measure and report the exosome concentration or at the minimum to discuss this limitation.

4.I did not see any additional figure regarding this or a modification that would clarify and ease the understanding of the primer strategy in details.

5.Thanks for validating this point to me.

6.Thanks for the information. I believe the readers would also benefit from the additional information provided in the rebuttal about the probe design.

7.Thanks. English edition mostly improved the manuscript.

Author Response

This manuscript is a resubmission of an earlier submission. The following is a list of the peer review reports and author responses from that submission.